# Multiple Futures Prediction

**Yichuan Charlie Tang**
Apple Inc.
yichuan_tang@apple.com

**Ruslan Salakhutdinov**
Apple Inc.
rsalakhutdinov@apple.com

## Abstract

Temporal prediction is critical for making intelligent and robust decisions in complex dynamic environments. Motion prediction needs to model the inherently uncertain future which often contains multiple potential outcomes, due to multi-agent interactions and the latent goals of others. Towards these goals, we introduce a probabilistic framework that efficiently learns latent variables to jointly model the multi-step future motions of agents in a scene. Our framework is data-driven and learns semantically meaningful latent variables to represent the multimodal future, without requiring explicit labels. Using a dynamic attention-based state encoder, we learn to encode the past as well as the *future* interactions among agents, efficiently scaling to any number of agents. Finally, our model can be used for planning via computing a conditional probability density over the trajectories of other agents given a hypothetical rollout of the 'self' agent. We demonstrate our algorithms by predicting vehicle trajectories of both simulated and real data, demonstrating the *state-of-the-art* results on several vehicle trajectory datasets.

## 1 Introduction

The ability to make good predictions lies at the heart of robust and safe decision making. It is especially critical to be able to predict the future motions of all relevant agents in complex and dynamic environments. For example, in the autonomous driving domain, motion prediction is central both to the ability to make high level decisions, such as when to perform maneuvers, as well as to low level path planning optimizations [34, 28].

Motion prediction is a challenging problem due to the various needs of a good predictive model. The varying objectives, goals, and behavioral characteristics of different agents can lead to multiple possible futures or modes. Agents' states do not evolve independently from one another, but rather they interact with each other. As an illustration, we provide some examples in Fig. 1. In Fig. 1(a), there are a few different possible futures for the blue vehicle approaching an intersection. It can either turn left, go straight, or turn right, forming different modes in trajectory space. In Fig. 1(b), interactions between the two vehicles during a merge scenario show that their trajectories influence each other, depending on who yields to whom. Besides multimodal interactions, prediction needs to scale efficiently with an arbitrary number of agents in a scene and take into account auxiliary and contextual information, such as map and road information. Additionally, the ability to measure uncertainty by computing probability over likely future trajectories of all agents in closed-form (as opposed to Monte Carlo sampling) is of practical importance.

Despite a large body of work in temporal motion predictions [24, 7, 13, 26, 16, 2, 30, 8, 39], existing state-of-the-art methods often only capture a subset of the aforementioned features. For example, algorithms are either deterministic, not multimodal, or do not fully capture both past and future interactions. Multimodal techniques often require the explicit labeling of modes prior to training. Models which perform joint prediction often assume the number of agents present to be fixed [36, 31].

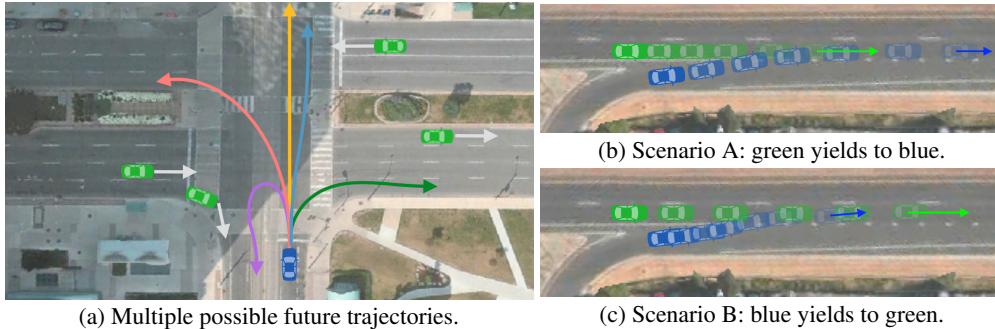

(a) Multiple possible future trajectories.

(b) Scenario A: green yields to blue.

(c) Scenario B: blue yields to green.

Figure 1: Examples illustrating the need for mutimodal interactive predictions. (a): There are a few possible modes for the blue vehicle. (b and c): Time-lapsed visualization of how interactions between agents influences each other's trajectories.

We tackle these challenges by proposing a unifying framework that captures all of the desirable features mentioned earlier. Our framework, which we call Multiple Futures Predictor (MFP), is a sequential probabilistic latent variable generative model that learns directly from multi-agent trajectory data. Training maximizes a variational lower bound on the log-likelihood of the data. MFP learns to model multimodal interactive futures *jointly* for all agents, while using a novel factorization technique to remain scalable to arbitrary number of agents. After training, MFP can compute both (un)conditional trajectory probabilities in closed form, not requiring any Monte Carlo sampling.

MFP builds on the Seq2seq [32], encoder-decoder framework by introducing latent variables and using a set of parallel RNNs (with shared weights) to represent the set of agents in a scene. Each RNN takes on the point-of-view of its agent and aggregates historical information for sequential temporal prediction for that agent. Discrete latent variables, one per RNN, automatically learn semantically meaningful modes to capture multimodality without explicit labeling. MFP can be further efficiently and jointly trained end-to-end for all agents in the scene. To summarize, we make the following contributions: First, semantically meaningful latent variables are automatically learned from trajectory data *without* labels. This addresses the **multimodality** problem. Second, interactive and parallel step-wise rollouts are preformed for all agents in the scene. This addresses the modeling of **interactions** between actors during future prediction, see Sec. 3.1. We further propose a dynamic attentional encoding which captures both the relationships between agents and the scene context, see Sec. 3.1. Finally, MFP is capable of performing hypothetical inference: evaluating the conditional probability of agents' trajectories conditioning on fixing one or more agent's trajectory, see Sec. 3.2.

## 2   Related Work

The problem of predicting future motion for dynamic agents has been well studied in the literature. The bulk of classical methods focus on using physics based dynamic or kinematic models [38, 21, 25]. These approaches include Kalman filters and maneuver based methods, which compute the future motion of agents by propagating their current state forward in time. While these methods perform well for short time horizons, longer horizons suffer due to the lack of interaction and context modeling.

The success of machine learning and deep learning ushered in a variety of data-driven recurrent neural network (RNN) based methods [24, 7, 13, 26, 16, 2]. These models often combine RNN variants, such as LSTMs or GRUs, with encoder-decoder architectures such as conditional variational autoencoders (CVAEs). These methods eschew physic based dynamic models in favor of learning generic sequential predictors (e.g. RNNs) directly from data. Converting raw input data to input features can also be learned, often by encoding rasterized inputs using CNNs [7, 13].

Methods that can learn multiple future modes have been proposed in [16, 24, 13]. However, [16] explicitly labels six maneuvers/modes and learn to separately classify these modes. [24, 13] do not require mode labeling but they also do not train in an end-to-end fashion by maximizing the data log-likelihood of the model. Most of the methods in literature encode the *past* interactions of agents in a scene, however prediction is often an independent rollout of a decoder RNN, independent of other future predicted trajectories [16, 29]. Encoding of spatial relationships is often done by placing other agents in a fixed and spatially discretized grid [16, 24].

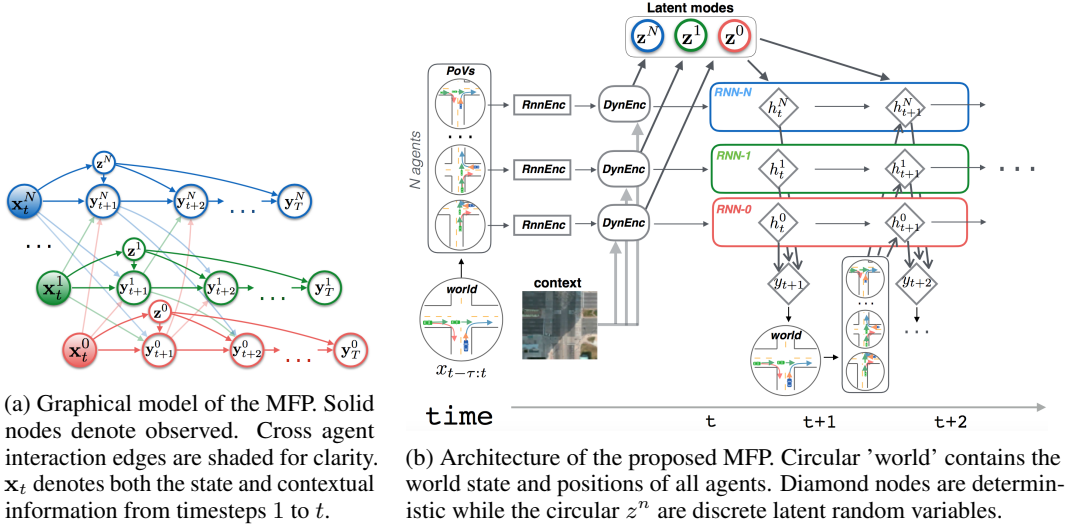

(a) Graphical model of the MFP. Solid nodes denote observed. Cross agent interaction edges are shaded for clarity. $\mathbf{x}_t$ denotes both the state and contextual information from timesteps 1 to $t$.

(b) Architecture of the proposed MFP. Circular 'world' contains the world state and positions of all agents. Diamond nodes are deterministic while the circular $z^n$ are discrete latent random variables.

Figure 2: Graphical model and computation graph of the MFP. See text for details. Best viewed in color.

In contrast, MFP proposes a unifying framework which exhibits the aforementioned features. To summarize, we present a feature comparison of MFP with some of the recent methods in the supplementary materials.

## 3 Multiple Futures Prediction

We tackle motion prediction by formulating a probabilistic framework of continuous space but discrete time system with a finite (but variable) number of $N$ interacting agents. We represent the joint state of all $N$ agents at time $t$ as $\mathbf{X}_t \in \mathbb{R}^{N \times d} \doteq \{\mathbf{x}_t^1, \mathbf{x}_t^2, \ldots, \mathbf{x}_t^N\}$, where $d$ is the dimensionality of each state[1], and $\mathbf{x}_t^n \in \mathbb{R}^d$ is the state $n$-th agent at time $t$. With a slight abuse of notation, we use superscripted $\mathbf{X}^n \doteq \{\mathbf{x}_{t-\tau}^n, \mathbf{x}_{t-\tau+1}^n, \ldots, \mathbf{x}_t^n\}$ to denote the past states of the $n$-th agent and $\mathbf{X} \doteq \mathbf{X}_{t-\tau:t}^{1:N}$ to denote the joint agent states from time $t - \tau$ to $t$, where $\tau$ is the past history steps. The future state at time $\delta$ of all agents is denoted by $\mathbf{Y}_\delta \doteq \{\mathbf{y}_\delta^1, \mathbf{y}_\delta^2, \ldots, \mathbf{y}_\delta^N\}$ and the future trajectory of agent $n$, from time $t$ to time $T$, is denoted by $\mathbf{Y}^n \doteq \{\mathbf{y}_t^n, \mathbf{y}_{t+1}^n, \ldots, \mathbf{y}_T^n\}$. $\mathbf{Y} \doteq \mathbf{Y}_{t:t+T}^{1:N}$ denotes the joint state of all agents for the future timesteps. Contextual scene information, e.g. a rasterized image $\mathbb{R}^{h \times w \times 3}$ of the map, could be useful by providing important cues. We use $\mathcal{I}_t$ to represent any contextual information at time $t$.

The goal of motion prediction is then to accurately model $p(\mathbf{Y}|\mathbf{X}, \mathcal{I}_t)$. As in most sequential modelling tasks, it is both inefficient and intractable to model $p(\mathbf{Y}|\mathbf{X}, \mathcal{I}_t)$ jointly. RNNs are typically employed to sequentially model the distribution in a cascade form. However, there are two major challenges specific to our multi-agent prediction framework: (1) **Multimodality**: optimizing vanilla RNNs via backpropagation through time will lead to *mode-averaging* since the mapping from $\mathbf{X}$ to $\mathbf{Y}$ is not a *function*, but rather a one-to-many mapping. In other words, multimodality means that for a given $\mathbf{X}$, there could be multiple distinctive modes that results in significant probability distribution over different sequences of $\mathbf{Y}$. (2) **Variable-Agents**: the number of agents $N$ is variable and *unknown*, and therefore we can not simply vectorize $\mathbf{X}_t$ as the input to a standard RNN at time $t$.

For multimodality, we introduce a set of stochastic latent variables $z^n \sim Multinoulli(K)$, one per agent, where $z^n$ can take on $K$ discrete values. The intuition here is that $z^n$ would learn to represent intentions (left/right/straight) and/or behavior modes (aggressive/conservative). Learning maximizes the marginalized distribution, where $z$ is free to learn any latent behavior so long as it helps to improve the data log-likelihood. Each $z$ is conditioned on $X$ at the current time (before future prediction) and will influence the distribution over future states $\mathbf{Y}$. A key feature of the MFP is that $z^n$ is only sampled once at time $t$, and must be consistent for the next $T$ time steps. Compared to sampling $z^n$ at every timestep, this leads to a tractability and more realistic intention/goal modeling,

as we will discuss in more detail later. We now arrive at the following distribution:

$$\log p(\mathbf{Y}|\mathbf{X}, \mathcal{I}) = \log(\sum_Z p(\mathbf{Y}, Z|\mathbf{X}, \mathcal{I})) = \log(\sum_Z p(\mathbf{Y}|Z, \mathbf{X}, \mathcal{I})p(Z|\mathbf{X}, \mathcal{I})), \qquad (1)$$

where $Z$ denotes the joint latent variables of all agents. Naïvely optimizing for Eq. 1 is prohibitively expensive and not scalable as the number of agents and timesteps may become large. In addition, the max number of possible modes is exponential: $\mathcal{O}(K^N)$. We first make the model more tractable by factorizing across time, followed by factorization across agents. The joint future distribution $\mathbf{Y}$ assumes the form of product of conditional distributions:

$$p(\mathbf{Y}|Z, \mathbf{X}, \mathcal{I}) = \prod_{\delta=t+1}^{T} p(\mathbf{Y}_\delta|\mathbf{Y}_{t:\delta-1}, Z, \mathbf{X}, \mathcal{I}), \qquad (2)$$

$$p(\mathbf{Y}_\delta|\mathbf{Y}_{t:\delta-1}, Z, \mathbf{X}, \mathcal{I}) = \prod_{n=1}^{N} p(\mathbf{y}_\delta^n|\mathbf{Y}_{t:\delta-1}, z^n, \mathbf{X}, \mathcal{I}). \qquad (3)$$

The second factorization is sensible as the factorial component is conditioning on the *joint* states of all agents in the immediate previous timestep, where the typical temporal delta is very short (e.g. 100ms). Also note that the future distribution of the $n$-th agent is explicitly dependent on its own mode $z^n$ but *implicitly* dependent on the latent modes of other agents by re-encoding the other agents predicted states $\mathbf{y}_\delta^m$ (please see discussion later and also Sec. 3.1). Explicitly conditioning an agent's own latent modes is both more scalable computationally as well as more realistic: agents in the real-world can only infer other agent's latent goals/intentions via observing their states. Finally our overall objective from Eq. 1 can be written as:

$$\log\big(\sum_Z p(\mathbf{Y}|Z, \mathbf{X}, \mathcal{I})p(Z|\mathbf{X}, \mathcal{I})\big) = \log\bigg(\sum_Z \prod_{\delta=t+1}^{T} \prod_{n=1}^{N} p(\mathbf{y}_\delta^n|\mathbf{Y}_{t:\delta-1}, z^n, \mathbf{X}, \mathcal{I})p(z^n|\mathbf{X}, \mathcal{I})\bigg) \qquad (4)$$

$$= \log\bigg(\sum_Z \prod_{n=1}^{N} p(z^n|\mathbf{X}, \mathcal{I}) \prod_{\delta=t+1}^{T} p(\mathbf{y}_\delta^n|\mathbf{Y}_{t:\delta-1}, z^n, \mathbf{X}, \mathcal{I})\bigg) \qquad (5)$$

The graphical model of the MFP is illustrated in Fig. 2a. While we show only three agents for simplicity, MFP can easily scale to any number of agents. Nonlinear interactions among agents makes $p(\mathbf{y}_\delta^n|\mathbf{Y}_{t:\delta-1}, \mathbf{X}, \mathcal{I})$ complicated to model. The class of recurrent neural networks are powerful and flexible models that can efficiently capture and represent long-term dependences in sequential data. At a high level, RNNs introduce deterministic hidden units $h_t$ at every timestep $t$, which act as features or embeddings that summarize all of the observations up until time $t$. At time step $t$, a RNN takes as its input the observation, $x_t$, and the previous hidden representation, $h_{t-1}$, and computes the update: $h_t = f_{rnn}(x_t, h_{t-1})$. The prediction $y_t$ is computed from the decoding layer of the RNN $y_t = f_{dec}(h_t)$. $f_{rnn}$ and $f_{dec}$ are recursively applied at every timestep of the sequence.

Fig. 2b shows the computation graph of the MFP. A *point-of-view* (PoV) transformation $\varphi^n(\mathbf{X}_t)$ is first used to transform the past states to each agent's own reference frame by translation and rotation such that $+x$-axis aligns with agent's heading. We then instantiate an encoding and a decoding RNN[2] per agent. Each encoding RNN is responsible for encoding the *past* observations $\mathbf{x}_{t-\tau:t}$ into a feature vector. Scene context is transformed via a convolutional neural network into its own feature. The features are combined via a *dynamic attention encoder*, detailed in Sec. 3.1, to provide inputs both to the latent variables as well as to the ensuing decoding RNNs. During predictive rollouts, the decoding RNN will predict its own agent's state at every timestep. The predictions will be aggregated and subsequently transformed via $\varphi^n(\cdot)$, providing inputs to every agent/RNN for the next timestep. Latent variables $Z$ provide extra inputs to the decoding RNNs to enable multimodality. Finally, the output $\mathbf{y}_t^n$ consists of a 5 dim vector governing a Bivariate Normal distribution: $\mu_x$, $\mu_y$, $\sigma_x$, $\sigma_y$, and correlation coefficient $\rho$.

While we instantiate two RNNs per agent, these RNNs *share* the same parameters across agents, which means we can efficiently perform joint predictions by combining inputs in a minibatch, allowing us to scale to *arbitrary* number of agents. Making $Z$ discrete and having only one set of latent variables influencing subsequent predictions is also a deliberate choice. We would like $Z$ to model modes generated due to high level intentions such as left/right lane changes or conservative/aggressive modes of agent behavior. These latent behavior modes also tend to stay consistent over the time horizon which is typical of motion prediction (e.g. 5 seconds).

**Learning**

Given a set of training trajectory data $\mathcal{D} = \{(\mathbf{X}^{(i)}, \mathbf{Y}^{(i)}, ) \ldots \}_{i=1,2,\ldots,|\mathcal{D}|}$, we optimize using the maximum likelihood estimation (MLE) to estimate the parameters $\theta^* = argmax_\theta \, \mathcal{L}(\theta, \mathcal{D})$ that achieves the maximum marginal data log-likelihood:[3]

$$\mathcal{L}(\theta, \mathcal{D}) = \log p(\mathbf{Y}|\mathbf{X}; \theta) = \log \Big( \sum_Z p(\mathbf{Y}, Z|\mathbf{X}; \theta) \Big) = \sum_Z p(Z|\mathbf{Y}, \mathbf{X}; \theta) \log \frac{p(\mathbf{Y}, Z|\mathbf{X}; \theta)}{p(Z|\mathbf{Y}, \mathbf{X}; \theta)} \quad (6)$$

Optimizing for Eq. 6 directly is non-trivial as the posterior distribution is not only hard to compute, but also varies with $\theta$. We can however decompose the log-likelihood into the sum of the *evidence lower bound* (ELBO) and the KL-divergence between the true posterior and an approximating posterior $q(Z)$ [27]:

$$\log p(\mathbf{Y}|\mathbf{X}; \theta) = \sum_Z q(Z|\mathbf{Y}, \mathbf{X}) \log \frac{p(\mathbf{Y}, Z|\mathbf{X}; \theta)}{q(Z|\mathbf{Y}, \mathbf{X})} + D_{KL}(q||p)$$

$$\geq \sum_Z q(Z|\mathbf{Y}, \mathbf{X}) \log p(\mathbf{Y}, Z|\mathbf{X}; \theta) + H(q), \quad (7)$$

where Jensen's inequality is used to arrive at the lower bound, $H$ is the entropy function and $D_{KL}(q||p)$ is the KL-divergence between the true and approximating posterior. We learn by maximizing the variational lower bound on the data log-likelihood by first using the true posterior[4] at the current $\theta'$ as the approximating posterior: $q(Z|\mathbf{Y}, \mathbf{X}) \doteq p(Z|\mathbf{Y}, \mathbf{X}; \theta')$. We can then fix the approximate posterior and optimize the model parameters for the following function:

$$Q(\theta, \theta') = \sum_Z p(Z|\mathbf{Y}, \mathbf{X}; \theta') \log p(\mathbf{Y}, Z|\mathbf{X}; \theta)$$

$$= \sum_Z p(Z|\mathbf{Y}, \mathbf{X}; \theta') \big\{ \log p(\mathbf{Y}|Z, \mathbf{X}; \theta_{rnn}) + \log p(Z|\mathbf{X}; \theta_Z) \big\}. \quad (8)$$

where $\theta = \{\theta_{rnn}, \theta_Z\}$ denote the parameters of the RNNs and the parameters of the network layers for predicting $Z$. As our latent variables $Z$ are discrete and have small cardinality (e.g. $< 10$), we can compute the posterior exactly for a given $\theta'$. The RNN parameter gradients are computed from $\partial Q(\theta, \theta')/\partial \theta_{rnn}$ and the gradient for $\theta_Z$ is $\partial KL(p(Z|\mathbf{Y}, \mathbf{X}; \theta')||p(Z|\mathbf{X}; \theta_Z))/\partial \theta_Z$.

Our learning algorithm is a form of the EM algorithm [14], where for the M-step we optimize RNN parameters using stochastic gradient descent. By integrating out the latent variable $Z$, MFP learns *directly* from trajectory data, without requiring any annotations or weak supervision for latent modes. We provide a detailed training algorithm pseudocode in the supplementary materials.

**Classmates-forcing**

*Teacher forcing* is a standard technique (albeit biased) to accelerate RNN and sequence-to-sequence training by using ground truth values $y_t$ as the input to step $t + 1$. Even with scheduled sampling [4], we found that over-fitting due to *exposure bias* could be an issue. Interestingly, an alternative is possible in the MFP: at time $t$, for agent $n$, the ground truth observations are used as inputs for all other agents $y_t^m : m \neq n$. However, for agent $n$ itself, we still use its previous predicted state instead of the true observations $x_t^n$ as its input. We provide empirical comparisons in Table 2.

**Connections to other Stochastic RNNs**

Various stochastic recurrent models in existing literature have been proposed: DRAW [20], STORN [3], VRNN [11], SRNN [18], Z-forcing [19], Graph-VRNN [31]. Beside the multi-agent modeling capability of the MFP, the key difference between these methods and MFP is that the other methods use continuous stochastic latent variables $z_t$ at *every* timestep, sampled from a standard Normal prior. The training is performed via the pathwise derivatives, or the reparameterization trick. Having multiple continuous stochastic variables means that the posterior can not be computed in closed form and Monte Carlo (or lower-variance MCMC estimators[5]) must be used to estimate the ELBO. This makes it hard to efficiently compute the log-probability of an arbitrary imagined or hypothetical trajectory, which might be useful for planning and decision-making (See Sec. 3.2). In contrast, latent variables in MFP is discrete and can learn semantically meaningful modes (Sec. 4.1).

With $K$ modes, it is possible to evaluate the exact log-likelihoods of trajectories in $\mathcal{O}(K)$, without resorting to sampling.

## 3.1 State Encodings

As shown in Fig. 2b, the input to the RNNs at step $t$ is first transformed via the *point-of-view* $\varphi(\mathbf{Y}_t)$ transformation, followed by *state encoding*, which aggregates the relative positions of other agents with respect to the $n$-th agent (*ego* agent, or the agent for which the RNN is predicting) and encodes the information into a feature vector. We denote the encoded feature $\mathbf{s}_t \leftarrow \phi_{enc}^n(\varphi(\mathbf{Y}_t))$. Here, we propose a dynamic attention-like mechanism where radial basis functions are used for matching and routing relevant agents from the input to the feature encoder, shown in Fig. 3.

Each agent uses a neural network to transform its state (positions, velocity, acceleration, and heading) into a key or descriptor, which is then matched via a radial basis function to a fixed number of "slots" with learned keys in the encoder network. The ego[6] agent has a separate slot to send its own state. Slots are aggregated and further transformed by a two layer encoder network, encoding a state $\mathbf{s}_t$ (e.g. 128 dim vector). The entire dynamic encoder can be learned in an end-to-end fashion. The key-matching is similar to dot-product attention [35], however, the use of radial basis functions allows us to learn spatially sensitive and meaningful keys to extract relevant agents. In addition, Softmax normalization in dot-product attention lacks the ability to differentiate between a *single* close-by agent vs. a far-away agent.

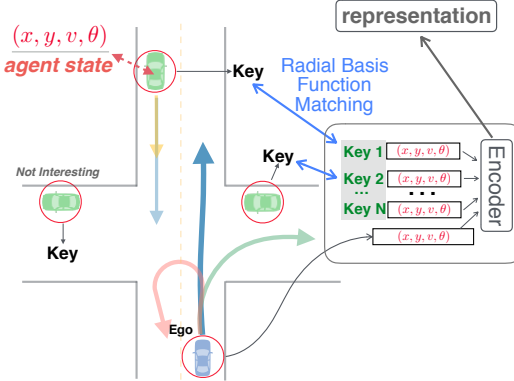

Figure 3: Diagram for dynamic attentional state encoding. MFP uses state encoding at every timestep to convert the state of surrounding agents into a feature vector for next-step prediction, see text for more details.

## 3.2 Hypothetical Rollouts

Planning and decision-making must rely on prediction for *what-ifs* [22]. It is important to predict how others might behave to different *hypothetical* ego actions (e.g. what if ego were to perform a more an aggressive lane change?). Specifically, we are interested in the distribution when conditioning on any hypothetical future trajectory $\mathbf{Y}^n$ of one (or more) agents:

$$p(\mathbf{Y}^{m:m\neq n}|\mathbf{Y}^n, \mathbf{X}) = \sum_{Z^{m:m\neq n}} \prod_{\delta=t+1}^{T} \prod_{m:m\neq n}^{N} p(\mathbf{y}_\delta^m|\mathbf{Y}_{t:\delta-1}, z^m, \mathbf{X})p(z^m|\mathbf{X}), \qquad (9)$$

This can be easily computed within MFP by fixing future states $\mathbf{y}_{t:T}^n$ of the conditioning agent on the R.H.S. of Eq. 9 while the states of other agents $\mathbf{y}_{t:T}^{m\neq n}$ are not changed. This is due to the fact that MFP performs interactive *future* rollouts in a synchronized manner for all agents, as the joint *predicted* states at $t$ of all agents are used as inputs for predicting the states at $t+1$. As a comparison, most of the other prediction algorithms perform independent rollouts, which makes it impossible to perform hypothetical rollouts as there is a lack of interactions during the future timesteps.

## 4 Experimental Results

We demonstrate the effectiveness of MFP in learning interactive multimodal predictions for the driving domain, where each agent is a vehicle. As a proof-of-concept, we first generate simulated trajectory data from the CARLA simulator [17], where we can specify the number of modes and script 2nd-order interactions. We demonstrate MFP can learn semantically meaningful latent modes to capture all of the modes of the data, all without using labeling of the latent modes. We then experiment on a widely known standard dataset of real vehicle trajectories, the NGSIM [12] dataset. We show that MFP achieves *state-of-the-art* results on modeling held-out test trajectories. In addition, we also benchmark MFP with previously published results on the more recent large scale Argoverse motion forecasting dataset [9]. We provide MFP architecture and learning details in the supplementary materials.

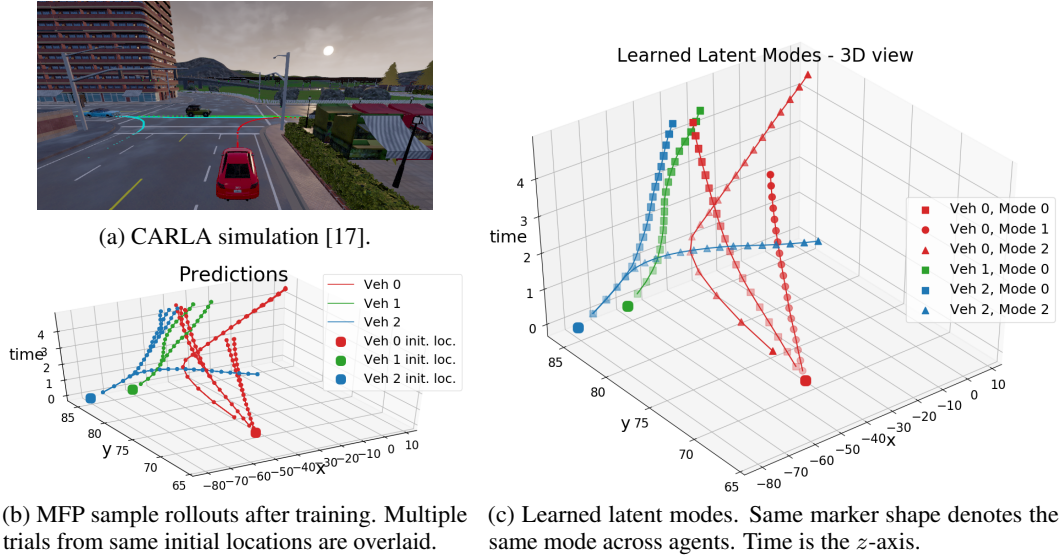

(a) CARLA simulation [17].

(b) MFP sample rollouts after training. Multiple trials from same initial locations are overlaid.

(c) Learned latent modes. Same marker shape denotes the same mode across agents. Time is the $z$-axis.

Figure 4: (a) CARLA data. (b) Sample rollouts overlayed, showing learned multimodality. (c) MFP learned semantically meaningful latent modes automatically: *triangle*: right turn, *square*: straight ahead, *circle*: stop.

## 4.1 CARLA

CARLA is a realistic, open-source, high fidelity driving simulator based on the Unreal Engine [17]. It currently contains six different towns and dozens of different vehicle assets. The simulation includes both highways and urban settings with traffic light intersections and four-way stops. Simple traffic law abiding "auto-pilot" CPU agents are also available.

We create a scenario at an intersection where one vehicle is approaching the intersection and two other vehicles are moving across horizontally (Fig. 4(a)). The first vehicle (red) has 3 different possibilities which are randomly chosen during data generation. The first mode aggressively speeds up and makes the right turn, cutting in front of the green vehicle. The second mode will still make the right turn, however it will slow down and yield to the green vehicle. For the third mode, the first vehicle will slow to a stop, yielding to both of the other vehicles. The far left vehicle also chooses randomly between going straight or turning right. We report the performance of MFP as a function of # of modes in Table 1.

| Metric (nats) | C.V. | RNN basic | MFP 1 mode | MFP 2 modes | MFP 3 modes | MFP 4 modes | MFP 5 modes |
|---|---|---|---|---|---|---|---|
| NLL | 11.46 | 5.64±0.02 | 5.23±0.01 | 3.37±0.81 | 1.72±0.19 | 1.39±0.01 | 1.39±0.01 |

Table 1: Test performance (minMSD with $K = 12$) comparisons.

| | Fixed-Encoding | DynEnc |
|---|---|---|
| NLL | 1.878±0.163 | 1.694±0.175 |

| | Teacher-forcing | Classmates-forcing |
|---|---|---|
| NLL | 4.344±0.059 | 4.196±0.075 |

Table 2: Additional comparisons.

| Metric | Vehicle 1 | | Vehicle 2 | |
|---|---|---|---|---|
| K=12 | Standard | Hypo | Standard | Hypo |
| minADE | $1.509 \pm 0.37$ | $1.402 \pm 0.34$ | $0.800 \pm 0.064$ | $0.709 \pm 0.060$ |
| minFDE | $2.530 \pm 0.635$ | $2.305 \pm 0.570$ | $3.171 \pm 0.462$ | $2.729 \pm 0.415$ |

Table 3: Hypothetical Rollouts.

The modes learned here are somewhat semantically meaningful. In Fig. 4(c), we can see that even for different vehicles, the same latent variable $z$ learned to be interpretable. Mode 0 (squares) learned to go straight, mode 1 (circles) learned to break/stop, and mode 2 (triangles) represents right turns. Finally, in Table 2, we can see the performance between using teacher-forcing vs. the proposed classmates-forcing. In addition, we compare different types of encodings. DynEnc is the encoding proposed in Sec. 3.1. Fixed-encoding uses a fixed ordering which is not ideal when there are $N$ arbitrary number of agents. We can also look at how well we can perform hypothetical rollouts by conditioning our predictions of other agents on ego's future trajectories. We report these results in Table 3.

|  | DESIRE [21] | SocialGAN | R2P2-MA [30] | ESP[30] no LIDAR | ESP | ESP Flex | MultiPath [8] | MFP-1 | MFP-2 | MFP-3 | MFP-4 | MFP-5 |
|---|---|---|---|---|---|---|---|---|---|---|---|---|
| Town01 test | 2.422 ±0.017 | 1.141 ±0.015 | 0.770 ±0.008 | 1.102 ±0.011 | 0.675 ±0.007 | 0.447 ±0.009 | 0.68 | 0.448 ±0.007 | 0.291 ±0.005 | **0.284** ±0.005 | **0.279** ±0.005 | 0.374 ±0.006 |
| Town02 test | 1.697 ±0.017 | 0.979 ±0.015 | 0.632 ±0.011 | 0.784 ±0.013 | 0.565 ±0.009 | 0.435 ±0.011 | 0.69 | 0.457 ±0.004 | 0.311 ±0.003 | 0.295 ±0.003 | **0.290** ±0.003 | 0.389 ±0.004 |

Table 4: Test performance (minMSD with $K = 12$) comparisons, in meters squared.

| Metric | time | Cons vel. | CVGMM[15] | [23] | MATF[39] | LSTM | S-LSTM[1] | CS-LSTM(M) | MFP-1 | MFP-2 | MFP-3 | MFP-4 | MFP-5 |
|---|---|---|---|---|---|---|---|---|---|---|---|---|---|
| NLL(nats) | 1 sec. | 3.72 | 2.02 | - | - | 1.17 | 1.01 | 0.89 (0.58) | 0.73±0.01 | -0.32±0.01 | -0.58±0.01 | **-0.65**±0.01 | -0.45±0.01 |
|  | 2 sec. | 5.37 | 3.63 | - | - | 2.85 | 2.49 | 2.43 (2.14) | 2.33±0.01 | 1.43±0.01 | 1.26±0.01 | **1.19**±0.01 | 1.36±0.01 |
|  | 3 sec. | 6.40 | 4.62 | - | - | 3.80 | 3.36 | 3.30 (3.03) | 3.17±0.01 | 2.45±0.01 | 2.32±0.01 | **2.28**±0.01 | 2.42±0.01 |
|  | 4 sec. | 7.16 | 5.35 | - | - | 4.48 | 4.01 | 3.97 (3.68) | 3.77±0.01 | 3.21±0.00 | 3.07±0.00 | **3.06**±0.00 | 3.17±0.00 |
|  | 5 sec. | 7.76 | 5.93 | - | - | 4.99 | 4.54 | 4.51 (4.22) | 4.26±0.00 | 3.81±0.00 | **3.69**±0.00 | **3.69**±0.00 | 3.76±0.00 |

| Metric | time | Cons vel. | CVGMM | | MATF | LSTM | S-LSTM | CS-LSTM[16] | MFP-1 | MFP-2 | MFP-3 | MFP-4 | MFP-5 |
|---|---|---|---|---|---|---|---|---|---|---|---|---|---|
| RMSE(m) | 1 sec. | 0.73 | 0.66 | 0.69 | 0.66 | 0.68 | 0.65 | 0.61 | **0.54**±0.00 | 0.55±0.00 | 0.54±0.00 | 0.54±0.00 | 0.55±0.00 |
|  | 2 sec. | 1.78 | 1.56 | 1.51 | 1.34 | 1.65 | 1.31 | 1.27 | **1.16**±0.00 | 1.18±0.00 | 1.17±0.00 | 1.16±0.00 | 1.18±0.00 |
|  | 3 sec. | 3.13 | 2.75 | 2.55 | 2.08 | 2.91 | 2.16 | 2.09 | **1.90**±0.00 | 1.92±0.00 | 1.91±0.00 | 1.89±0.00 | 1.92±0.00 |
|  | 4 sec. | 4.78 | 4.24 | 3.65 | 2.97 | 4.46 | 3.25 | 3.10 | **2.78**±0.00 | 2.80±0.00 | 2.78±0.00 | 2.75±0.00 | 2.78±0.00 |
|  | 5 sec. | 6.68 | 5.99 | 4.71 | 4.13 | 6.27 | 4.55 | 4.37 | **3.83**±0.01 | 3.85±0.01 | 3.83±0.01 | 3.78±0.01 | 3.80±0.01 |

Table 5: NGSIM prediction results. Hightlighted columns are our results (lower is better). MFP-$K$: $K$ is the number of latent modes. The standard error of the mean is over 5 trials. For multimodal MFPs, we report minRMSE over 5 samples. NLL can be negative as we are modeling a continuous density function.

## CARLA PRECOG

We next compared MFP to a much larger CARLA dataset with published benchmark results. This dataset consists of over 60K training sequences collected from two different towns in CARLA [30]. We trained MFP (with 1 to 5 modes) on the Town01 training set for 200K updates, with minibatch size 8. We report the minMSD metric (in meters squared) at $\hat{m}_{K=12}$ for all 5 agents *jointly*. We compare with state-of-the-art methods in Table 4. Non-MFP results are reported from [30] (v3) and [8]. MFP significantly outperforms various other methods on this dataset. We include qualitative visualizations of test set predictions in the supplementary materials.

## 4.2 NGSIM

Next Generation Simulation [12](NGSIM) is a collection of video-transcribed datasets of vehicle trajectories on US-101, Lankershim Blvd. in Los Angeles, I-80 in Emeryville, CA, and Peachtree St. in Atlanta, Georgia. In total, it contains approximately 45 minutes of vehicle trajectory data at 10 Hz and consisting of diverse interactions among cars, trucks, buses, and motorcycles in congested flow.

We experiment with the US-101 and I-80 datasets, and follow the experimental protocol of [16], where the datasets are split into 70% training, 10% validation, and 20% testing. We extract 8 seconds trajectories, using the first 3 seconds as history to predict 5 seconds into the future.

In Table 5, we report both neg. log-likelihood and RMSE errors on the test set. RMSE and other measures such as average/final displacement errors (ADE/FDE) are not good metrics for multimodal distributions and are only reported for MFP-1. For multimodal MFPs, we report minRMSE over 5 samples, which uses the ground truth select the best trajectory and therefore could be overly optimistic. Note that this applies equally to other popular metrics such as minADE, minFDE, and minMSD.

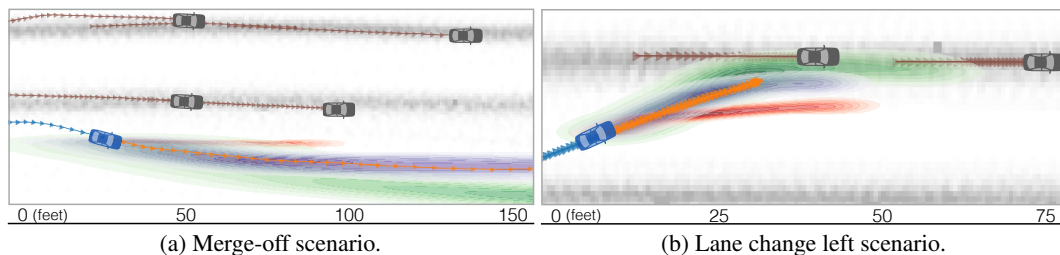

| (a) Merge-off scenario. | (b) Lane change left scenario. |

Figure 5: Qualitative MFP-3 results after training on NGSIM data. Three modes: red, purple, and green are shown as density contour plots for the blue vehicle. Grey vehicles are other agents. Blue path is past trajectory, orange path is actual future ground truth. Grey pixels form a heatmap of frequently visited paths. Additional visualizations provided in the supplementary materials.

The current state-of-the-art, multimodal CS-LSTM [16], requires a separate prediction of 6 fixed maneuver modes. As a comparison, MFP achieves significant improvements with less number of modes. Detailed evaluation protocols are provided in the supplementary materials. We also provide qualitative results on the different modes learned by MFP in Fig. 5. In the right panel, we can interpret the green mode is fairly aggressive lane change while the purple and red mode is more "cautious". Ablative studies showing the contributions of both interactive rollouts and dynamic attention encoding are also provided in the supplementary materials. We obtain best performance with the combination of both interactive rollouts and dynamic attention encoding.

### 4.3 Argoverse Motion Forecasting

Argoverse motion forecasting dataset is a large scale trajectory prediction dataset with more than $300,000$ curated scenarios [9]. Each sequence is 5 seconds long in total and the task is to predict the next 3 seconds after observing 2 seconds of history.

We performed preliminary experiments by training a MFP with 3 modes for 20K updates and compared to the existing official baselines in Table 6. MFP hyperparmeters were not selected for this dataset so we do expect to see improved MFP performances with additional tuning. We report validation set performance on both version 1.0 and version 1.1 of the dataset.

| minADE K=6 | C.V. | NN+map | LSTM+ED | LSTM ED+map | MFP3 (ver. 1.0) | MFP3 (ver. 1.1) |
|---|---|---|---|---|---|---|
| meters | 3.55 | 2.28 | 2.27 | 2.25 | 1.411 | 1.399 |

Table 6: Argoverse Motion Forecasting. Performance on the validation set. CV: constant velocity. Baseline results are from [9].

### 4.4 Planning and Decision Making

The original intuitive motivation for learning a good predictor is to enable robust decision making. We now test this by creating a simple yet non-trivial reinforcement learning (RL) task in the form of an unprotected left turn. Situated in Town05 of the CARLA simulator, the objective is to safely perform an unprotected (no traffic lights) turn, see Fig. 6. Two oncoming vehicles have random initial speeds. Collisions incur a penalty of $-500$ while success yields $+10$. There is also a small reward for higher velocity and the action space is acceleration along the ego agent's default path (blue).

Using predictions to learn the policy is in the domain of model-based RL [33, 37]. Here, MFP can be used in several ways: 1) we can generate imagined future rollouts and add them to the experiences from which temporal difference methods learns [33], or 2) we can perform online planning by using a form of the shooting methods [5], which allows us to optimize over future trajectories. We perform experiments with the latter technique where we progressively train MFP to predict the joint future trajectories of all three vehicles in the scene. We find the optimal policy by leveraging the current MFP model and optimize over ego's future actions. We compare this approach to a couple of strong model-free RL baselines: DDPG and Proximal policy gradients. In Fig. 7, we plot the reward vs. the number of environmental steps taken. In Table 7, we show that MFP based planning is more robust to parameter variations in the testing environment.

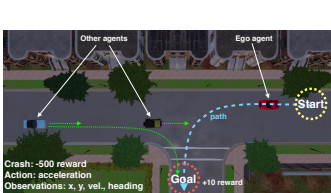

Figure 6: RL learning environment - Unprotected left turn.

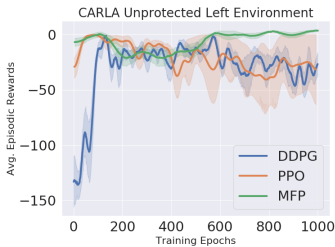

Figure 7: Learning curves as a function of step sizes.

| $\Delta$ Env. Params | DDPG | PPO | MFP |
|---|---|---|---|
| $vel : +0m/s$ | 3% | 4% | 0% |
| $vel : +5m/s$ | 8% | 4% | 0% |
| $vel : +10m/s$ | 6% | 15% | 0% |
| $acc : +1m/s^2$ | 3% | 1% | 0% |

Table 7: Testing crash rates per 100 trials. Test env. modifies the velocity & acceleration parameters.

## 5 Discussions

In this paper, we proposed a probabilistic latent variable framework that facilitates the joint multi-step temporal prediction of arbitrary number of agents in a scene. Leveraging the ability to learn latent modes directly from data and interactively rolling out the future with different point-of-view encoding, MFP demonstrated *state-of-the-art* performance on several vehicle trajectory datasets. For future work, it would be interesting to add a mix of discrete and continuous latent variables as well as train and validate on pedestrian or bicycle trajectory datasets.

**Acknowledgements** We thank Barry Theobald, Gabe Hoffmann, Alex Druinsky, Nitish Srivastava, Russ Webb, and the anonymous reviewers for making this a better manuscript. We also thank the authors of [16] for open sourcing their code and dataset.

## Footnotes

[1]We assume states are fully observable and are agents' $(x, y)$ coordinates on the ground plane ($d$=2).

[2]We use GRUs [10]. LSTMs and GRUs perform similarly, but GRUs were slightly faster computationally.

[3]We have omitted the dependence on context $\mathcal{I}$ for clarity. The R.H.S. is derived from the common *log-derivative trick*.

[4]The ELBO is the tightest when the KL-divergence is zero and the $q$ is the true posterior.

[5]Even with IWAE [6], 50 samples are needed to obtain a somewhat tight lower-bound, making it prohibitively expensive to compute good log-densities for these stochastic RNNs for online applications.

[6]We will use ego to refer to the main or 'self' agent for whom we are predicting.

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
