[Supplementary Material · mfp_supp.pdf]

# Multiple Futures Prediction - Supplementary Materials

**Yichuan Charlie Tang**
yichuan_tang@apple.com

**Ruslan Salakhutdinov**
rsalakhutdinov@apple.com

## 1 MFP Implementation Details

Figure 1: MFP Architecture.

### 1.1 Architecture

For all of our experiments with use the same model architecture for the MFP. The input observations consists of 2 dimensional $x$, $y$, positions of the agent. Both encoders and decoders are using bidirectional GRUs. The encoder RNNs have a hidden dimension of 64 while the decoder RNNs have a hidden layer size of 128. For nonlinear activation functions we use the rectified linear (ReLU) activations.

**Context**: Context information is encoded by two layers of standard $3 \times 3$ convolutional layers followed by one 2d max-pooling layer, followed by one more $3 \times 3$ convolutional and finally a fully connected layer with an output dimensionality of 64.

**Dynamic encoding**: For dynamic encoding we use a small neural network of 32 units each to transform every agent state (position) into an 8 dimensional key. The encoder has 8 slots, where each slots has a learnable key. Matching is performed using Radial Basis function $RBF(k, k') = \exp(-\frac{||k-k'||_2^2}{T})$, where $T$ is the temperature and set to $1.0$. Match score are used as part of a soft-attention mechanism to weight the contribution of each encoder RNN features and aggregated into

the 8 slots. The input vector of the 8 slots are transformed by a 3 hidden layer network into a feature encoding of size 32.

**Latent modes**: A single linear layer is used to project the concatenated feature encodings into a Softmax which represent the distribution over the $K$ modes. The input dimension consists of $64 + 32 + 32 = 128$ dimensions and the output dimensions is $K$.

**Decoder**: The decoder RNNs consist of bi-directional GRUs with 128 dimensional hidden layers and are initialized with the contextual features at timestep $t$.

**Normalization**: We found that training is accelerated by computing the average future trajectory positions (constant across samples) and subtracting it off from the future prediction targets. In addition, we normalize the past historical trajectories by subtracting off the position of the agents at time $t$, such that the position of each agent at time $t$ for its own RNN would be $(0.0, 0.0)$.

## 1.2 Algorithm

We provide the pseudo-code to MFP training in the Algorithm 1.

---

**Algorithm 1:** MFP training algorithm

---

**Input :** Dataset $\mathcal{D} = \{(\mathbf{X}^{(i)}, \mathbf{Y}^{(i)}, ) \dots \}_{i=1,2,\dots,|\mathcal{D}|}$, # of modes M
**Initialize :** Randomly initialize MFP network, encoders, decoders, and all parameters.

1   **for** *iterations $i = 0$ to $T$* **do**
2     Randomly sample a neighborhood trajectory cluster from $\mathcal{D}$.
3     Normalization of trajectories.
     `// forward pass`
4     **for** *agent $i$ in cluster* **do**
5       $\mathbf{h}_i \leftarrow$ GRU encoder of past history of agent $i$
6     **end**
7     $\mathcal{I} \leftarrow$ Contextual encoding of the scene.
8     Dynamic Attention Encoding (Sec. 3.1): $\mathbf{f} \leftarrow DynEnc(\{\mathbf{h}_i\}, \mathcal{I})$.
9     **for** *modes $m = 1$ to $M$* **do**
10       **for** *steps $\delta = t + 1$ to $T$* **do**
11         **for** *agent $i$ in cluster* **do**
12           Set $z^i$ according to mode $m$ (one-hot encoding).
13           $\mathbf{e}_i \leftarrow DynEnc(\varphi(\hat{\mathbf{Y}}_{\delta-1}))$ `// DynEnc of past state of all agents with agent` $i$`'s point-of-view`
14           For classmates-forcing use ground truth $\mathbf{y}_{\delta-1}^n$ when $n \neq i$.
15           $\hat{\mathbf{y}}_\delta^i \leftarrow RnnDecode(\mathbf{e}_i, z^i, \mathbf{f})$
16         **end**
17         Update predicted $\hat{\mathbf{Y}}_\delta = \{\hat{\mathbf{y}}_\delta^i\}_{i=0:N}$
18       **end**
19     **end**
20     Compute loss.
     `/* E-step                                                    */`
21     Compute true posterior $p(\mathbf{Z}|\mathbf{Y}, \mathbf{X}; \theta)$
     `/* M-step                                                    */`
22     Backpropagation through time over entire forward computation graph (weighted by the posterior).
23     Update approximating prior: $\partial KL(p(Z|\mathbf{Y}, \mathbf{X}; \theta')||p(Z|\mathbf{X}; \theta_Z))/\partial \theta_Z$
24     Update the all parameters via ADAM optimizer.
25 **end**

---

## 1.3 Feature Comparisons

We provide a comparison of our proposed MFP and some of the existing prior work on trajectory predictions in the table below. "Variational" approaches maximize a variational lower-bound during optimization. "Interactive rollouts" means that the future predicted states of one agent will interact with the future predictions of other agents. "Hypothetical" refers to whether a model is able to change its prediction depending on a hypothetical ego future trajectory.

Table 1: *Comparison to existing work on vehicle trajectory predictions.*

| Methods | Multimodal | No-labeling of Modes | Variational | Inter. Rollouts | Inter. Encodings | Context | Hypothetical |
|---|---|---|---|---|---|---|---|
| CS-LSTM | ✔ | ✕ | ✕ | ✕ | ✔ | ✕ | ✕ |
| Seq2Seq | ✕ | - | ✕ | ✕ | ✔ | ✕ | ✕ |
| TrafficPredict | ✕ | - | ✕ | ✔ | ✔ | ✕ | ✕ |
| Cui et al. (Uber) | ✔ | ✔ | ✕ | ✕ | ✔ | ✔ | ✕ |
| DESIRE | ✔ | ✔ | ✔ | ✕ | ✔ | ✔ | ✕ |
| IntentNet | ✔ | ✕ | ✕ | ✕ | ✔ | ✔ | ✕ |
| ChaufferNet | ✕ | - | ✕ | ✕ | ✔ | ✔ | ✕ |
| MFP (Ours) | ✔ | ✔ | ✔ | ✔ | ✔ | ✔ | ✔ |

## 2  Carla Experiments

For CARLA experiments, we collected trajectory data for 3 vehicles in an intersection from the map *Town-05*. We generate data at 20 Hz by simulating forward for 5 seconds. The position of vehicles are initialized at the same location with a Gaussian noise of $\pm 0.2$ meters. The red vehicle in Fig. 4(a) has 3 different modes: *right turn fast*, *right turn slow*, *stop/yield*, which are selected at random. The far-left blue vehicle can also go straight or turn right, chosen randomly. We generate 100 trajectories from random simulations in total.

### 2.1  Interactions

Figure 2: The same mode results in different behaviors due to interactions. Orange trajectory for left and right rollouts are from the same mode. However, as the blue trajectory changes, the interaction affects the rollout of the orange trajectory (vehicle 1).

Figure 3: The same mode results in different behaviors due to interactions. Green and orange trajectory for left and right rollouts are from the same mode. However, as the blue trajectory changes, the interaction affects the rollout of both vehicles 1 and 2.

One interesting properties of the interactive future rollout of the MFP is that interaction allows the same mode to take on multiple trajectories. As an example, vehicle 1 might be in an aggressive mode, but depending on whether or not another vehicle cut in front of vehicle 1, it will never-the-less have a different trajectory. This means that having interactions in *future* joint rollouts allows MFP to model

more variation with less latent modes. We illustrate this with a couple of examples in Fig. 2 and Fig. 3.

## 2.2 Carla PRECOG

We qualitatively visualize prediction results of MFP-4 on the test sets of Town01 and Town02 in Figs. 4 and 5. We overlay predictions on top of Lidar point clouds. Predictions are very accurate and the trajectories do not seem to contain more than two modes, as can also be seen in the quantitative experiments. In this case, MFP's attention-based state encoding becomes critical to learning accurate predictions. We also note that the quantitative minMSD results have improved significantly from the previous version of this paper. The main differences are due to the increase of the size of the minibatch from 1 to 8 during training and better data normalization.

(a) Example 1.

(b) Example 2.

(c) Example 3.

(d) Example 4.

Figure 4: Qualitative joint predictions from Town01 test set.

## 3 NGSIM Experiments

For NGSIM experiments, we used 6 sequences from the US-101 and I-80 datasets and randomly split them into 70% training, 10% validation, and 20% testing, resulting in 5922867 samples for training, 859769 samples for validation, and 1505756 samples for testing. Following practice of existing work, we use 3 seconds for past history while predicting the next 5 seconds into the future. We perform subsampling by a factor of 2 so every timestep is 200 ms.

(a) Example 1.

(b) Example 2.

(c) Example 3.

(d) Example 4.

Figure 5: Qualitative joint predictions from Town02 test set.

Figure 6: NGSIM dataset.

Figure 7: MFP ablative studies.

Results from Table 6 are over 5 random trials where we report the standard error of the mean. For all MFP models we reported errors using the saved model parameters after 300K parameter updates.

The dataset is openly available here: `https://data.transportation.gov/Automobiles/ Next-Generation-Simulation-NGSIM-Vehicle-Trajector/8ect-6jqj`.

### 3.1 Training

For training, we use the ADAM optimizer with stochastic minibatches of trajectory samples. For the NGSIM experiments, the minibatch size varies depending on the density of vehicles around a particular region. The average minibatch size is 60, but minibatch size of $1$ or $\geq 200$ are also possible. The initial learning rate is set at 0.001 and are decreased by a factor of 10 every 100K updates. We lower-bound the learning rate to be 0.00005. In order to accelerate training, we first pretrain MFP without interactive rollouts for 200K updates before continuing to train with interactive rollouts for an additional 100K parameter updates.

### 3.2 Runtime

For NGSIM experiments, a training run is performed using a single nVidia Titan X GPU using the pyTorch 1.0.1 framework. During training, each batch update takes roughly 1.2 seconds wall clock time. Each training run converges in approximately one day.

### 3.3 Ablative Studies

In Figure 7 we show the effects of interactions and dynamic encoding in NGSIM test performance. Best results are obtained with the utilization of both.

#### 3.3.1 Additional Qualitative Experiments

We show additional qualitative experiments similar to the ones shown in Figure 5. The colors represent the three modes learned by MFP-3: red, purple, and green. Blue and brown paths are previous trajectories while the orange path is the future trajectory.

(a) Example 1.

(b) Example 2.

(c) Example 3.

(d) Example 4.

Figure 8: *Additional qualitative plots showing MFP-3 modes learned from NGSIM dataset.*