[Reviews · NeurIPS 2019]

Reviewer 1



Updated review: Authors responded to most of the concerns and hence increasing my score by one. Please include the additional experiments performed after submission that was included in rebuttal in the final copy if the paper was accepted. --- Quality: High, it’s a well-written paper justifying and then describing the model step by step and testing it on one simulated and one real dataset. Clarity: Medium. Most of the paper is clear and easy to follow. The learning section could be improved by perhaps describing the algorithm in an algorithm box. Significance: low: It’s a bit hard to evaluate the significance as the architecture seems to specifically designed for the driving domain and there is no discussion how this could be used or generalized in other applications. -The assumption of small cardinality of latent code and its discrete nature is a strong one. A discussion on how limiting this could be and potential solutions to extend this model would be useful. -L172: Could you expand here, as what would be the E step. -Fig 2-a: typo: y_[t+1]

Reviewer 2



>Originality: Q: Are the tasks or methods new? A: The task is not new, the method is new. Q: Is the work a novel combination of well-known techniques? A: The work is a novel combination of well-known techniques. Q: Is it clear how this work differs from previous contributions? A: Yes. Q: Is related work adequately cited? A: Mostly, except one very related work is missing. >Quality: Q: Is the submission technically sound? A: I think so. Q: Are claims well supported by theoretical analysis or experimental results? A: Most of the claims are supported by evidence, however there is no evaluation of the claimed contribution of "hypothetical inference". Q: Is this a complete piece of work or work in progress? A: It is a work in progress: there are some important missing experiments. Q: Are the authors careful and honest about evaluating both the strengths and weaknesses of their work? A: It seems so, except there's no clear attempt made to analyze weaknesses of the approach. >Clarity: Q: Is the submission clearly written? A: The submission is mostly clear, although there are some technical ambiguities. Q: Is it well organized? (If not, please make constructive suggestions for improving its clarity.) A: Yes. Q: Does it adequately inform the reader? (Note: a superbly written paper provides enough information for an expert reader to reproduce its results.) A: I think some readers might be able to implement a version of this approach. The training algorithm is a complex mixture of many different components. >Significance: Q: Are the results important? A: The results indicate the model is a feasible approach for modeling multi-agent vehicle motion. Q: Are others (researchers or practitioners) likely to use the ideas or build on them? A: I think the graphical model presented holds promise for others to extend. Q: Does the submission address a difficult task in a better way than previous work? A: The approach has strengths with respect to previous work (tractable graphical model allows for exact inference) but they're accompanied by weaknesses (training algorithm is complex). Q: Does it advance the state of the art in a demonstrable way? A: It demonstrably advances the state-of-the-art on the NGSIM dataset, which is somewhat simplistic given its simple context. Q: Does it provide unique data, unique conclusions about existing data, or a unique theoretical or experimental approach? A: It sounds like new data (on CARLA) will be released. Below are my major comments / concerns (unordered): -- It's unclear what some of the columns in Table 1 mean. Does context mean including past trajectories, image/LIDAR information, or both? (After reading further beyond the table, it appears to be both, but the Table should be clear when it is presented) Isn't DESIRE [21] variational, since it uses a cVAE? (Related: why is "variational" an important attribute of the model to list in the table?). Does "inter-rollouts" mean "interacting rollouts"? What about "inter. encodings"? What is "hypothetical"? This table is almost more confusing than enlightening. The meaning of the attributes should be clear, and each attribute should be meaningful. -- Equation 5, 6 derivations are unclear (although I am fairly sure they're correct). I didn't follow the derivation of (5), at least not by using the mentioned log derivative trick. The derivation of Eq6 RHS was cryptic as well; I didn't see where Jensen's could be applied. However (6) is derived, please make sure the derivation is clear, and put in appendix if space doesn't allow. Here's my derivation of the RHS of (6) directly, which might be simpler (dropping X, \theta dependence), as it doesn't rely on Jensen's or the log-derivative trick. log p(y) = sum_z q(z|y) log p(y) = sum_z q(z|y) log [ p(y,z)/p(z|y) ] = sum_z q(z|y) log [ p(y,z) ] + H(q,p) = sum_z q(z|y) log [ p(y,z) ] + KL(q,p) + H(q) >= sum_z q(z|y) log [ p(y,z) ] + H(q) (since KL >= 0) -- L193 "possible to evaluate the exact log-likelihoods of trajectories" It would be helpful here to clarify that [if I'm correct] something like Eq(4) is used to evaluate log-likelihoods. It's not clear what precise units and coordinate system NLL is calculated in (cross-entropy depends on the coordinate system of the trajectories). Can the exact NLL metric be given analytically? Is the NLL computed for each agent's trajectory individually, or the joint (I'm assuming the joint)? Is there some normalization going on in the metric (e.g. by number of agents or by timesteps)? -- There's some related uncited work that I think the authors should be aware of [A]. Although [A] is too recent to expect quantitative comparison (since it appeared within a month of the beginning of the NeurIPS submission deadline), the work shares many similarities that merit discussion, as it overlaps with the submission's stated contributions: 1) interative and parallel step-wise rollouts for all agents 2) ability to perform "hypothetical inference" / planning. [A] demonstrated using their graphical model and likelihood function to condition on goal states of one agent, and plan the latent variables to infer how other agents might respond, and showed that this inference can improve forecasting performance. -- There should be some sample quality metric on the CARLA data. The common one (referred to as minADE or minMSD) is computed by fixing a budge of samples, e.g. 10 or 20, and reporting the smalled MSD to the ground-truth sample. -- There's no comparison to any other state-of-the-art methods on the collected CARLA dataset. That makes the CARLA experiments much less informative w.r.t. to prior work. -- "Hypothetical inference" was stated as a capability of the model, but there's no qualitative or quantitative results that provide evidence of this capability. -- Are the latent modes for each agent fairly robust to variation in scene context and the latent modes for the other agents? There's only a single qualitative example that the modes are semantically meaningful (Fig4c). Is there a metric (aside from human verification) that could be used to roughly measure whether the meaning of the modes retains semantic meaning across scenes and variation in other agents? If this claim isn't that integral to the paper, it could be removed. -- It's unclear what the context is in the NGSIM experiments. Is the visual context used at all? Is it just past trajectory context? Below are my minor comments / concerns (unordered): The title is not very informative -- it's hard to predict much about the contents of the paper (even the data domain!) from the title. Please make it more informative. It's not clear 1) that the problem is multi-agent 2) the domain is contextual vehicle motion prediction. L105 I'm not familiar with a "Discrete(K)" distribution. Perhaps the authors meant "Categorical(K)". Minor suggestion, use "\big(" and "\big)" for outer parentheses, e.g. after the second equality in (5). L171 Is it really a KL? The non-RNN terms of Eq(7) RHS look like a cross-entropy -- isn't it d/\theta_z H(p(z|y,x;\theta') , p(z|x;\theta_z) ? The T and Hz of the modelled trajectories in CARLA are not clear. [A] "PRECOG: PREdiction Conditioned On Goals in Visual Multi-Agent Settings." Rhinehart et al. arXiv preprint arXiv:1905.01296

Reviewer 3



Review of "Multiple Futures Prediction" Summary They propose a method for predicting multiple possible rollouts in a stochastic dynamic environment. Their approach involves extending seq2seq to have a population of RNN agents (i.e. teacher-forcing => classmates-forcing), so it is able to predict multimodal states. In addition, each RNN can represent a particular agent in the environment, and implicitly model agent-to-agent interactions and incorporate other agent behavior directly in future predictions. They demonstrate SOTA results on NGSIM dataset, and also experiment their approach on data collected from CARLA simulation environment. I thought the paper was well written; the text is clear and well reasoned, and it feels the authors have put effort into making polished visualizations, diagrams and figures. I particularly liked the effort that also went into the supplementary materials to ensure details are described to facilitate reproduction (the qualitative results were also interesting). Existing work specific to vehicle trajectory prediction is described. They obtained good results of the benchmark dataset in their domain. I think this work should be accepted, however I would like to suggest a few improvements that will make the paper much more solid for NeurIPS, if the authors want to try to improve a bit. Right now I feel the paper is at a level where I can assign a score of 6 (will explain the points I think is lacking below), but with a bit more work, it can easily be a paper with a score of 7 or 8 from me (see below). The motivation behind this paper, as quoted from the abstract, is "Temporal prediction is critical for making intelligent and robust decisions in complex dynamic environments" and clearly the approach should look to find use cased in decision-making applications (i.e. an agent acting an a real or simulation environment that has complicated dynamics), so the thing I found lacking in the paper was that all the experiments were only based on how well the approach can be used to model a pre-recorded dataset in log-likelihood estimation metrics. The experiments are important, don't get me wrong, especially the fact that they can compare to a large body of existing work using NGSIM, but I think for this paper to have a much higher impact beyond niche (i.e. to the larger NeurIPS community), the authors should consider applying their method to actual tasks where they can measure performance in terms of rewards, task objects, etc. Given that the authors have used the CARLA simulator (for the purpose of collecting a dataset, and fitting their model to that dataset only), one can consider using CARLA environment to define actual difficult tasks (it is an RL environment afterall), and seeing how the approach facilitates an agent to solve difficult, but important tasks that self-driving agents need to tackle. With that being said, in the RL literature, there is an existing body of work (though may not deal directly with autonomous driving domain) that also tackles the problems motivated by "Motion prediction needs to model the inherently uncertain future which often contains multiple potential outcomes, due to multi-agent interactions and the latent goals of others." For instance, some examples in this line of work in RL [1, 2, 3] proposed methods to build generative latent models of the environment from collected data, and sample multi-model tragectories of the future, fits a model to collected dataset of experiences, and more importantly address the (IMO) more critical question of how to use these models to actually solve difficult problems. Would be nice to see a discussion that links the line of work of this paper, to the literature in probabilistic generative models used in RL domain that has similar goals and motivation. I believe this will make the work more impactful, since readers can find ideas from other subfields and be inspired to creatively devise more innovative solutions down the road to tackle these important problems. Overall, I like this paper and the proposed approach, and think this is a thoughtful and encouraging work. I'm recommending acceptance (with a score of 6 for now), but if the authors address some of these concerns (esp if they can get results on RL tasks, and report performance on the actual task, rather than log likelihood), then I should be able to improve the score by 1-2 points. [1] Racanière and Weber and Reichert et al., Imagination-Augmented Agents for Deep Reinforcement Learning, NIPS2017. https://arxiv.org/abs/1707.06203 [2] Ha and Schmidhuber, Recurrent World Models Facilitate Policy Evolution, NeurIPS2018. https://arxiv.org/abs/1809.01999 [3] Hafner et al., Learning Latent Dynamics for Planning from Pixels, ICML2019 (arxiv, Nov 2018). https://arxiv.org/abs/1811.04551

Reviewer 4



Summary: The paper proposes a novel method for predicting future trajectories of multiple vehicles in the scene. The approach is validated on two datasets (CARLA and NGSIM) and sets the new state-of-the art on NGSIM. The scores below are consistent with "overall score" legend. Originality: 6. 1) The approach is novel. It relies on RNNs and graphical models to estimate the probabilities of future trajectories. A novel algorithm similar to EM is proposed for training the model. 2) The approach differs from previous work in the way that multiple possible futures are represented. 3) The related work is reasonably cited. However, this paper also seems very related, is there a reason not to mention it? "Multi-Modal Trajectory Prediction of Surrounding Vehicles with Maneuver based LSTMs" https://arxiv.org/abs/1805.05499 Quality: 6. The datasets and baselines are reasonably diverse. The experiments seem well-executed. Clarity: 3. The reviewer found it hard to understand the core ideas of the paper and follow the mathematical derivations. Comments: 1) The notation seems confusing. For example, in L88-89 X^n corresponds to the data up to the time step t, however, there is no t in the notation. The same is true for Y^n. 2) How exactly does the point-of-view transformation (L136) work? Why is it needed? 3) How are Z variables handled exactly in the optimization code? Are they sampled at some point? How does NLL computation work? 4) Why is the model with 1 latent mode (MFP-1) better than CS-LSTM? That is, if there is a single mode, what is the purpose of latent variable modeling? Minor comments: 1) In Table 1, why are "Variational" and "Hypothetical" columns important? What does "Hypothetical" mean? 2) L283: compare -> compared 3) L302: in interesting -> interesting Significance: 5. Given the current explanation and the absence of code, it could be hard to implement the method and reproduce the results. -------------------------------------------------------------------------- -------------------------------------------------------------------------- POST-REBUTTAL UPDATE: Given the body of additional experiments, the promise to open-source the code and clarify the explanations, I am increasing my score from 5 to 6. The method seems to work sufficiently better than the baselines (even though it's quite heavy for understanding, in my opinion) and the presence of code should make it reproducible. If the paper is accepted, I would strongly suggest that the comments mentioned above in the "Clarity" section are addressed in the camera-ready version of the paper. It would also be ideal if the code is open-sourced by the camera-ready deadline.

[Author Response · NeurIPS 2019]

We thank the reviewers for valuable feedback and will make the suggested changes. We've included additional experiments to address the
existing concerns/areas of improvements. Reviewer 2: sec. C provides additional results on non-vehicle classes (i.e. bike & ped.) Reviewer 3:
here we compare MFP directly to PRECOG[A] on their released CARLA data in Tab. 1. MFP *significantly* outperforms previous SOTA in [A]
for 5 agent joint predictions. We also quantitatively evaluated *hypothetical* inference in Tab. 2. We report new results using the minMSD
sample metric. Reviewer 5: in sec. B, we created a CARLA-based RL env. and task and proposes a simple MPF-based shooting policy (a form
of MPC). We compared it with several SOTA model-free methods, demonstrating faster training and leading to a safer or more robust policy.
Reviewer 6: We will release code in the near future and make the suggested clarifications. Please see detailed responses below.

**REV2**: The learning algorithm box was included in the Appendix. The E-step computes the true posterior distribution (step 21 of Alg. 1). MFP
is a general framework and does not assume any vehicle specific dynamics or priors. MFP can perform multi-agent joint predictions of any
objects, not just vehicles. (e.g. N-body problem or physics-based interactions). Sec. C. shows new results on bike and ped. prediction tasks.

**REV3**: We will clarify Tab.1 in the paper and add derivations of Eqs. 5, 6. DESIRE[21] is variational. Variational learning (e.g. EM or
VAEs) is principled with certain guarantees on the ELBO. "inter." means interactive and hypothetical means the ability to perform conditional
inference by fixing a particular agent's future trajectory. NLL is computed in closed form and is normalized by the num of timesteps and
agents. For NGSIM the coordinate is in feet. Ablative studies to test how latent modes change could be performed by removing certain agents
from the scene. NGSIM results did not use visual context, as visual context (grey lines of Fig.5) did not improve performance in a significant
way. MFP of Tab. 1 below use 100x100x4 LIDAR rasterization as visual context. Our CARLA trajectories are 5 seconds at 20 Hz. The KL
term (L171) differs from cross-entropy by a constant term (not dependent on $\theta_Z$), so we can use them interchangeably. We compare with
[A]'s CARLA dataset and MFP achieves new ***state-of-the-art*** on the most challenging 5-agents joint prediction task in Town02. [A] is closely
related to MFP but one difference is that MFP can handle arbitrary number of agents while [A] (if we're not mistaken) requires a fixed num of
$N$ agents. We will certainly cite and compare/contrast with [A], and we thank the authors of [A] for providing their dataset.

**REV5**: We thank the reviewer for a thoughtful review and one of the original motivations was better decision making. We will add discussions
to the referred RL papers. MFP can be used to learn better $p(s'|s,a)$ for model-based RL. In sec. B, we connect predictions to RL by creating
a hard self-driving RL task in CARLA. We use MFP to learn good predictive models and then our policy (a form of MPC) can use Shooting
methods[H] to check for future collisions within $\tau$ meters. We show episodic reward curves and also test for robustness/safety by changing the
distribution of initial conditions of other agents. We achieve superior performance compare to SOTA model-free Deep RL methods.

**REV6**: The mentioned "Multi-modal ..." paper is a previous version of the CSLSTM paper [8] which we cite and compare with. PoV
normalization rotates and translate the observations of other agents to an ego-centric frame and helps learning. Z variables are enumerated
during the E-step. NLL is simply neg. log-likelihood and can be computed in closed form. MFP-1 is better than CSLSTM due our dynamic
attention mechanism. MFP-1 is just a baseline to compare to other unimodal methods. Hypothetical refers to Sec 3.2 and variational learning is
desirable as it is probabilistically sound. We will clarify these points and open source our code in the near future.

**A. COMPARISON TO PRECOG [A] (CARLA).** We train MFP (with and without LIDAR; 3, 5, and 7 modes) on the PRECOG CARLA
dataset [A]. MFP is trained on 60,701 Town01 sequences for 300K updates. We report apples-to-apples comparison using minMSD metric
$\hat{m}_{K=12}$ on Town02 testset for all 5 agents *jointly*. MFP (green) achieve SOTA results in Tab. 1. A quantitative eval of sec. 3.2. is in Tab. 2.

**Table 1:** *CARLA (PRECOG) Town02.* minMSD computed exactly as Eq. 13 of [A].

| minMSD (meters) | DESIRE [21] | SocialGAN [C] | R2P2-MA [D] | ESP[A] no LIDAR | MFP5 no LIDAR | ESP[A] | MFP3 | MFP5 | MFP7 |
|---|---|---|---|---|---|---|---|---|---|
| 5 agents *joint* $\hat{m}_{K=12}$ | 2.422 ±0.017 | 1.141 ±0.015 | 0.770 ±0.008 | 1.102 ±0.011 | 0.842 ±0.025 | 0.675 ±0.007 | 0.641 ±0.018 | 0.553 ±0.013 | **0.496** **±0.011** |

**Table 2:** *Hypothetical Rollouts.* Ex. from Fig. 4(a).

| $\hat{m}_{K=10}$ | MFP3 | | MFP3+Hypothetical | |
|---|---|---|---|---|
| (meters) | Veh1(blue) | Veh2(green) | Veh1(blue) | Veh2(green) |
| minMSD | 2.081 ± 0.25 | 2.765 ± 0.18 | **1.764 ± 0.13** | **2.199 ± 0.14** |
| minFDE | 3.137 ± 0.18 | 3.419 ± 0.31 | **2.732 ± 0.12** | **2.742 ± 0.26** |

**B. CARLA RL ENVIRONMENT - UNPROTECTED LEFT TURN.** We create an unprotected left turn task in CARLA Town05, where
the objective is for Ego to safely complete an unprotected (no traffic lights) turn. Two oncoming vehicles have random initial speeds. MFP
can be used in model-based RL in multiple ways: first is similar to Dyna-Q[I], where a MFP can be used to generate imagination rollouts
to be added to the experience buffer. The second is an online planning algorithm (Shooting), where Ego's future action sequences are
optimized to maximize for the planning reward under the learned MFP dynamics model. We compare this with several SOTA model-free
methods and show that MFP-Shooting requires less sample complexity and is more robust to variations in test environment parameters.

**Table 3:** *Testing crash rates per 100 trials. Test env. modifies the velocity & acceleration of other vehicles to test for generalization.*

| $\Delta$ Env. Params | DDPG[E] | PPO2[F] | C51[G] | MFP-S $\tau$=5m | MFP-S $\tau$=10m |
|---|---|---|---|---|---|
| $vel : +0m/s$ | 2% | 1% | 0% | 0% | 0% |
| $vel : +5m/s$ | 7% | 4% | 5% | 1% | 0% |
| $vel : +10m/s$ | 13% | 5% | 7% | 0% | 0% |
| $acc : +1m/s^2$ | 9% | 3% | 2% | 1% | 0% |

**C. BIKE AND PEDESTRIAN PREDICTIONS.** We perform additional experiments on the Stanford Drone Dataset (SDD) [J] for ped.
and bike predictions. We train MFP on videos 0,1,2,4,5 of the *deathCircle* scene and test on video3. Red and blue lines are the mean predicted
trajectory of two modes and the green is the predicted multi-modal log-probability density. MFP performs significantly better than baselines.

**Figure 1:** *Left: predicted bike trajs. Right: selected future prob. density for bikes.*

**Table 4:** *SDD: bike and ped. predictions on 'deathCircle' scene, video 3. Past:3 secs. Future:5 secs.* $\hat{m}_{K=12}$.

| | metric | Cons Vel. | RNN | CSLSTM | MFP3 | MFP5 |
|---|---|---|---|---|---|---|
| Bike | minMSD(pixels) | 10.31 | 8.75 | 8.44 | 5.34 | 4.77 |
| | Neg. LL(nats) | - | 5.67 | 5.25 | 2.03 | 1.74 |
| Ped. | minMSD(pixels) | 4.33 | 3.28 | 3.01 | 2.61 | 2.14 |
| | Neg. LL(nats) | - | 3.39 | 3.07 | 1.44 | 1.31 |

**References**: **[A]** *PRECOG, Rhinehart et al. ICCV '19.* **[C]** *SocialGAN, Gupta et al. CVPR '18.* **[D]** *R2P2, Rhinehart et al. ECCV '18.* **[E]** *Deep DPG, Lillicrap et al. '15.*
**[F]** *Proximal Policy Optimization, Schulman et al. '18. https://github.com/openai/baselines.* **[G]** *Dopamine: Castro et al. '18. https://github.com/google/dopamine.* **[H]** *Robust*
*Constrained MPC, Richards A. G. Phd Thesis, '05.* **[I]** *Dyna an Integrated Architecture, Sutton '91.* **[J]** *Stanford Drone Dataset, Robicquet et al., ECCV '16.*


[Meta-Review · NeurIPS 2019]

The recommendations were initially split. Following the authors' rebuttal, the negative reviewers revised their recommendations upward in the discussion, although there are still multiple concerns, as expressed in the reviews. The authors are requested to thoroughly address the reviewers' concerns in the revision.